# Mobile-Application Loading-Animation Design and Implementation Optimization

Jesenka Pibernik *, Jurica Dolić, Lidija Mandić and Valentina Kovač

Department of Graphic Design and Imaging, University of Zagreb Faculty of Graphic Arts, 10000 Zagreb, Croatia
* Correspondence: jpiberni@grf.hr; Tel.: +385-996067669

**Featured Application: Research activities from the fields of human–computer interaction, psychology/neuroscience, and programming have addressed understanding user wait-time experience and perceived loading-animation performance. Analysis of the experimental results has shown which features of loading-animation design directly impact the user's quality of experience. The issues surrounding the implementation of design decisions related to perceived load speed metrics are discussed.**

**Abstract:** Mobile-application performance and ultimately bounce rate are influenced by many factors, with wait-time duration proven to be the most important influencing factor. It is therefore important to optimize loading-time perception and performance to ensure a smooth and enjoyable experience. This research investigates the influences of loading-animation design, motion speed, and semantics, as well as the graphics format used for creation, on time perception. These research aims were (1) to identify the key factors that influence the quality of the user experience in the design of animated loading indicatorsand their perceived dimensions, (2) to find out what impact animation meaning has on perceived time duration, and (3) to identify loading-animation-design decisions' influence on application performance. Research activities from the fields of human–computer interaction, psychology/neuroscience, and networking/telecommunications have addressed understanding the user's wait-time experience. In this experiment, we confirmed that the perceived wait-time duration of a loading indicator is positively related to nontemporal properties, such as the processing principle, but we also reviewed the influence of the semantic structure of perceptual time representation. The specific meaning of an animation metaphor and its narrative sequence is related to information extraction and processing of novel and repeated stimuli and, consequently, to the experienced delight and duration judgment of an end user.

**Keywords:** loading animation; user experience; wait-time duration; semantic structure; file formats





## 1. Introduction

An application's loading time is important and rated highly in the user-experience (UX) hierarchy. In general, users' behavior changes after 2 s of waiting for simple information-retrieval tasks [1]. This is not surprising, as nothing can happen until a page is loaded, or at least assumed to be loaded. The specific impact of loading time on bounce rate depends on a variety of factors, including the user's expectations, the complexity of the application, and the performance of the device on which the app is running. A slow response could be caused by the network condition, the number of concurrent users, or the operation itself. For example, if the loading animation uses a lot of CPU or GPU memory resources, it can slow down performance. Perceived loading speed is a measure of how quickly application loading appears to the user. It is important because it can have a significant impact on user experience, even if the actual loading time is relatively fast [2]. Therefore, real-world performance (a statistically measurable metric) and perceived performance (a qualitative measurement of efficiency) differ. Researchers should focus on user-centric performance

metrics that can influence the user's loading experience, rather than measuring loading with just one metric [3,4].

Depending on how a loading animation is implemented, it can have an impact on the overall performance of an application. Loading animations are often created using computer-generated or computer-assisted 2D/3D animations in Adobe After Effects and coded in CSS, JavaScript, or Scalable Vector Graphics (SVG). To ensure smooth motion, developers must ensure that each frame renders in less than 16 ms, even if the device's natural frame rate is 60 frames per second (which is faster than the user can perceive). For this reason, rendering in the SVG format may require some extra XML code writing and optimization. Recently, a JSON-based animation file format called Lottie was introduced. One of the major benefits of Lottie, compared to XML formats, is its small file size, which can significantly improve download speed and reduce disk-space usage. In addition, designers who use Adobe After Effects can immediately start creating Lottie content and preview and optimize animations on iOS, Android, and React Native without any coding. In other words, using JSON may simplify and improve the workflow of designers and developers compared to the workflow that is created with Extensible Markup Language family file format [5].

We can conclude that visual design principles are at the core of good performance in mobile applications [6]. From this viewpoint, speed is becoming a dimension of the design process. Through making a loading screen less generic and more novel, designers can make time seem to pass more quickly and the wait more pleasant [7,8]. On the other hand, as the complexity of a loading animation increases, the file size of the loading format also increases. This makes reducing file size an important consideration for successful mobile-application development, as it can improve download speed and reduce the amount of disk space used without sacrificing aesthetics [9]. Therefore, it is important to optimize loading-animation design and implementation to ensure a smooth and enjoyable user experience.

## 1.1. Time Affordance

User-experience researchers are concerned with different and strongly interrelated factors that may influence quality of experience, encompassing human, system, and context influence factors [10,11]. In the following, we will group and discuss a few main factors, along with their interplay, in the context of the loading-animation experience. It has been proven by many practitioners and researchers in the field of human–computer interaction that a wait-indicator animation can mask slow system-loading performance and even increase perceived performance through decreased perceived wait time. Loading animation may be defined as a subtype of progress animation, as it informs the user that the process of loading is active. Focus on the animated wait indicator maintains a consistent visual experience and conveys the sense of a performant page load. For that purpose, researchers have introduced a concept of time affordance and a set of principles for determining whether the diagnostic information available to the user is rich enough to prevent unproductive and even destructive actions due to an unclear understanding of progress [12]. For instance, Nielsen Norman recommends using a progress-indicator animation for any action that takes longer than about 1.0 s.

Since loading animation is a visual representation of temporal change, its cognitive and affective functions are there to inform users of changes in the current operational state. Therefore, there are a lot of animation and implementation techniques that are not directly related to performance but help create smoother and more fluid motion [13]. They might even help improve the perception of performance as well. Temporal animations can make wait times shorter through offering people valuable content, such as tips, quotes, etc., to make the wait time more meaningful [14]. If the loader's animation and design are customized and intended to metaphorically provoke a meaning, that meaning usually indicates future actions, product functions, or brand identification. Metaphors provide cues to users about how to understand products: to orient and personify. In this article, we will measure the effectiveness of such metaphors, compared to more abstract content, in

shortening of time perception. In addition, we will tackle recent development of animation-coding techniques.

### 1.2. Subjective Duration of Time

For the aim of distinguishing the most important research aspects of the subjective experience of duration, an overview of the theoretical background will be described. The present paragraph aims to distinguish the main issues related to users' explicit judgments about length estimation and quality of experience of temporal intervals. This analysis will be limited to research that has linked visual perception to the perception of time. People do not perceive time itself, but changes in or passage of time, or what might be described as "events in time". The perceived time interval between two successive events is referred to as the perceived duration [15]. Perception of duration requires a minimum of about 0.1 s in the case of visual stimuli, such as a flash. Stimuli of any shorter time than this are therefore perceived as instantaneous and not representative of any duration at all. Salti et al. defined subjective perception as a dynamic updating process in which all stimuli are coordinated [16]. That model assumes that time perception, like any other form of conscious perception, is governed by stimulus saliency, the observer's goal and motivation, and context. In our study, the user's goal and the context, e.g., a mobile application, will remain the same, so stimulus attributes will be discussed.

Subjective duration of time studies has been conducted in a cross-disciplinary manner through different experimental methods, explained with different theoretical models, and linked to different perceptual and cognitive processes, many of these with contradictory results. We will focus on research that concerns two different groups of studies: psychology/neuroscience and human–computer interaction.

### 1.3. Time Perception in Psychology and Neuroscience

Understanding of temporal processing of intervals between 2 and 5 s, which require the support of cognitive resources, can reveal further paradigms that affect duration estimates. The findings of a duration-judgment experiment in which the participants were asked to make judgments after each interval presentation in the form of a moving visual stimulus helped us define variables for further research. The first link was identified between temporal evaluation and visio–motor representation of motor actions. Most importantly, the causes of distortions of perceived duration were explained with mentally simulated movement. In other words, visual representation of movement might interfere with the user's subjective time-estimation process, leading to some degree of time distortion.

Through introduction of the concept of subjectively experienced time (SXT), user-experience researchers connected time estimation and appraisal theory of emotion to time perception [17,18]. They conveyed the idea that the subjective feeling of time passing is more important than any objective measure of duration. SXT is individual and context-dependent and can be influenced by several factors, including past experiences and present cognitive and affective state. The psychological appraisal theory of emotion describes how people judge the pleasantness of an experience after that experience, based on their interpretation of the situation [19,20]. In other words, appraisal is necessary for understanding how we feel about the outcome of time estimation. Liikkanen and Gómez' research has proven that experiences are evaluated positively when the passing of time is not noticed, or at least when no waiting is perceived.

Findings have also confirmed the influences of different affective states on wait-animation-design perception [21]. Since emotions are momentary reactions that are oriented toward specific objects or events, they are highly influential on animation evaluation. At its core, an emotional experience can be characterized as a feeling of satisfaction or dissatisfaction. More recent findings have also reported the impact of affective valance (positive and negative) on perceived time judgments according to duration (2, 4, or 6 s) [22].

Most researchers agree that human time awareness is, along with other bodily states, related to some aspects of attention [23], memory, fixation, and emotion [24–27]. Attention is

a cognitive process of selective concentration on certain external objects (visual or auditory) while paying less or no attention to others. It has been proven that level of attentional demand is strongly linked to perceived duration. Previous research has confirmed shorter duration judgment for repeated compared to novel stimuli [28]. Those findings are in line with the "processing principle", which attributes lengthening of time intervals to perceptual vividness and ease of extracting information from stimuli. In other words, more exposure to the same stimuli reduces the number of resources allocated to perceptual processing. Here, a distinction must be made between conceptual and perceptual fluency: in other words, between ease of retrieval and general processing fluency [29]. Perceptual fluency is also linked to higher aesthetic judgments and more enjoyable experiences.

Researchers have concluded that duration judgments seem to be a reliable way to assess cognitive load, with cognitive load being a measure of mental effort experienced by an end user, which is directly in line with the widely accepted definition of QoE [30,31]. With a higher cognitive load, the prospective-duration-judgment ratio (subjective duration to objective duration) decreases, but the retrospective ratio increases. Prospective and retrospective paradigms differ from each other depending on whether the subject knows the importance of the passage of time in a given task. However, whether viewing fewer or less-complex stimuli requires a lower cognitive load remained unclear.

### 1.4. Time Perception in Human–Computer Interaction

Loading-animation designers' concern is guidance of user focus and visual attention as a limited cognitive ability to select stimuli from a screen for some period. Visual attention, both automatic and conscious, is attracted to shapes, sizes, colors, and movement of graphical elements from which data is collected. UX designers use motion to grab users' spatial- and feature-based attention because they are familiar with the phenomenon that the more conscious users are, the longer they experience wait time. The processes of attentional orientation and saccades distort perceived duration [32,33]. According to findings, through offloading of the interpretation of changes of an application operation to the perceptual system, animation allows a user to continue thinking about the task domain with no need to shift the context to the interface domain [34,35]. Those findings are in line with psychological studies that have indicated that more changes, greater complexity, and a larger number of reaches in each situation are characterized by vivid stimuli and elicit a notion of time passing more quickly [36].

However, there is still a gap in understanding what type of loading animation has the best results in terms of both perceived duration and measured user-experience quality. Major operating systems' design guidelines (Material Design and iOS) specify two main, visually distinct types of progress animations: looped animation (circular) and progress bar indicators (linear). Although widely accepted, their usage is not without negative connotations. According to Luke Wroblewski, using spinners has its downside: "It's like watching the clock tick down—when you do, time seems to go slower" [37]. Studies have confirmed that visually augmented and determined progress bars could be used to make processes appear faster [38,39].

A semantic approach to describe and understand form, movement and meaning in loading animation has not yet been modeled. Some research works address the semantic framework for understanding what kind of loading-animation message influences perceived performance and user experience [40]. Viget's experiment confirmed that branded loaders hold participants on the loading page longer and have lower abandon rates than do nonbranded, generic loaders [41]. Because they are based on expectations, prior knowledge, and past experiences, animation metaphor semantics resist all forms of modeling [42,43]. The standard measure for the interpretation effectiveness of a proposed loading metaphor has not been yet established. Traditionally, this has been studied with several subjective measures, including the semantic differential [44], user preference [45], time perception, satisfaction, acceptability, and appropriateness.

## 2. Materials and Methods

After analysis of the complex interplay of psychophysical perception of motion and subjectively experienced time, we extracted other important experience parameters that influence wait time perception and user satisfaction. This research was planned as subjective duration judgment and quality assessment based on psycho–visual experiments that represent the fundamental and most reliable ways to assess users' QoE, focusing on the impact of temporal factors (e.g., wait times) on end users' QoE assessments.

Subjective duration and passage of time are commonly measured using psychophysical tasks. Based on previous research mentioned earlier, prospective/retrospective timing has a key influence on duration judgment. Since our experiment was based on an experimental within-group factorial design, the retrospective measure was not an option. In our experiments, time-interval duration was fixed across trials; only the stimulus (the type of loading animation) varied. Participants classified each stimulus on a 5-point scale, ranging from long/short and slow/fast, and provided a numerical estimate of its duration in seconds. For each task, the mean judgment assigned to a given stimulus provided a measure of its apparent speed and duration. The numerical estimate was compared to the actual duration of each animation [46].

Even though speed is rated highest in the user experience (UX) hierarchy, the aesthetics and semantics of loading animations are important for QoE as well, since they are a part of the basis for achieving higher user delight. The choice of parameters in Experiment I was related to the way users perceive, retrieve, and process information from loading animations. Therefore, perceptual fluency and ease of extracting information were measured with aesthetic judgment, emotional and functional impact with the semantic differential, and cognitive load and stimulus novelty (processing fluency) through interest. The impact of visual attention was determined with eye motion in Experiment II. An eye-tracking tool was the key objective method to measure attention.

The choices of stimuli were based on user-experience experiments that employed visually and semantically distinct types of loading animations, e.g., loop, progress bar, identity animation, etc. Each type differed in terms of animation-graphics style: its novelty, meaning, level of abstraction, and relation to the application's functions. Frame rate and timing functions were applied consistently. Four out of eight animations were based on repetitive spinning: the most common type of loading animation. Three animations had novel and narrative content related to the application's function (coffee ordering). We assumed that adding meaning and novelty to an animation would result in a higher-quality user experience (Figure 1).

### 2.1. Experiment 1—Perception Measure

These assessments were conducted in a laboratory setting (an artificial and controlled research environment to isolate and investigate the impacts of specific factors). In this research, influence factors (IFs) were dependent variables, whereas the resulting experience, as perceived by the end user (feature), was an independent variable [47].

### 2.1.1. Participants

A total of 30 participants, students who volunteered for this research, were tested; 13 of them fit into the age category of 18 to 24; 12 fit into the age category of 25 to 44, and 5 fit into the age category of above 45. Informed consent was given, and participants could discontinue this experiment without prior notice. All had normal or corrected-to-normal vision; 26% of participants were male, while 74% were female. A chi-square test proved that there was no statistical difference between the theoretical and empirical values for age and gender in this group of participants.

### 2.1.2. Stimuli and Apparatus

For this research experiment, eight different prototypes related to the mobile food-and-drink application titled Coffee2go were created. Based on an analysis of 33 categories

of apps in Google Play, our designer created eight different loading animations that represented the main types currently used for mobile applications. All of the animations fit the application's function, color scheme, and design style (Table 1). The wait-animation speed was set at 24 frames per second and the duration was set to 5.0 s each. All of the animations had standard easing (accelerate at the beginning and decelerate at the end). Besides four abstract, repetitive geometrical forms, such as circular and linear progress bars (animations 1, 3, 5, and 6), four more-complex "storytelling" animations were designed to provoke semantical meaning related to the identity (animation 2), attributes (animation 4), activity (animation 7), and motivation (animation 8) associated with the business that the application Coffee2go supports.

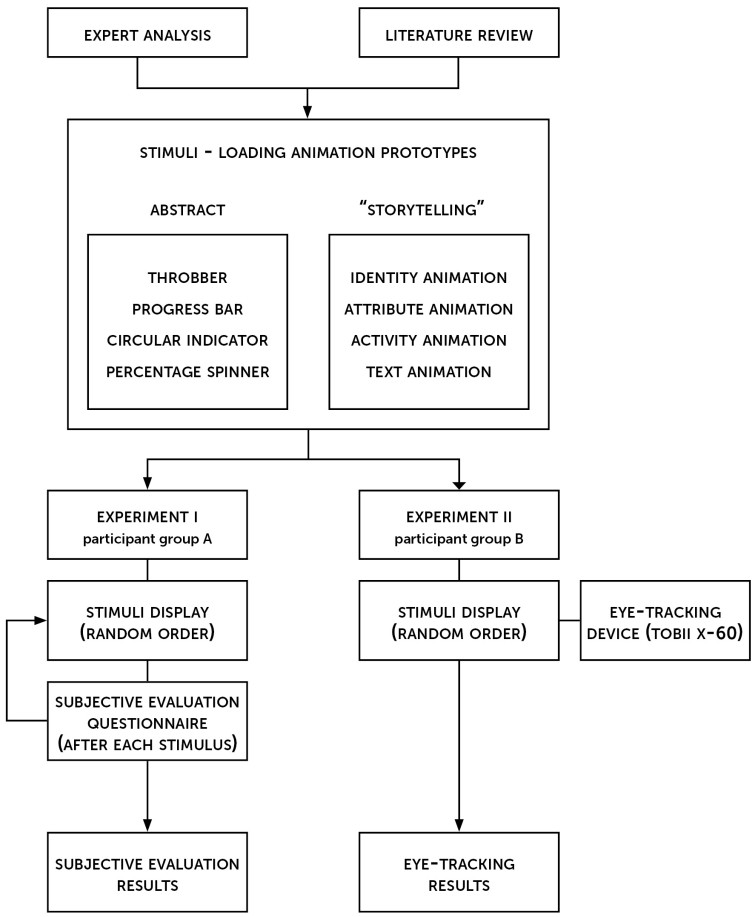

**Figure 1.** Flow chart of research process and methodology.

Based on the popular app's mean loading time, Vierordt's law, and Nielsen Norman recommendations, each wait-time animation was set to five seconds [48,49]. Based on the Weber–Fechner Law [50,51] and the just-noticeable-difference 20% rule, we predicted that each user's perceived duration should be within the margin of 1 s. Therefore, we offered participants to choose a duration from 3.0, 3.5, 4.0, 4.5, 5.0, 5.5, and 6.0 s.

### 2.1.3. Procedure

Eight application prototypes, which differed only in loading-animation design, were saved on a Samsung Galaxy mobile phone, and participants perceived the application to be "real". Tests were provided individually, and all eight prototypes were shown to each participant in random order. Participants were asked to focus on time perception while waiting for the application to load (prospective paradigm) [52].

**Table 1.** Loading-animation type and design descriptions.

| | Loading Animation | Animation Metaphor | Time Affordance |
|---|---|---|---|
| 1. | Throbber | Spinning icon (repetitive) | Indeterminate |
| 2. | Identity Animation | The app logo on a cup, which is being filled with coffee, in front of a coffee machine | Anticipative |
| 3. | Progress Bar (With Percentage Indicator) | A linear graphical control element with a percentage indicator | Determinate |
| 4. | Attribute Animation | Icons with text (coffee, grounded, brewed, served) shown in the linear sequence | Indeterminate |
| 5. | Circular Indicator | Twelve radial lines arranged in a circular pattern (repetitive) | Indeterminate |
| 6. | Percentage Spinner | "Spinning wheel" with a percentage indicator (repetitive) | Determinate |
| 7. | Activity Animation | Cup being filled with coffee from a coffee jug | Anticipative |
| 8. | Text Animation | "do not let anything stop you", "skip obstacles", "take it with you", "go for your own Coffee2go o" | Indeterminate |

Each participant participated in the experiment under the supervision of an investigator at the User Experience Lab at the Faculty of Graphic Arts, University of Zagreb, in a lightly obscured area with constant light conditions of neutral coloration. The session began with a brief introduction to the application's functionality and completion of background data. The participants declared that they liked coffee and were each promised to receive a cup of coffee at the end of the experiment.

The mobile phone was handed to each participant with the splash screen opened, and the participants were asked to continue using the application. After the loading animation was over and the transition to the home screen ended, the participant was asked to complete a survey. Including a "think time" in performance tests makes performance research more realistic, as it represents users' actual behavior in a system more accurately. Think time is the time when a real user waits between actions. It is defined as the time between the completion of one request and the start of the next request.

### 2.1.4. Results with Discussion

An analysis of the collected data was conducted using statistical methods (descriptive analysis, independent samples t-test, and one-way repeated measures MANOVA). The descriptive statistics for the QoE features are presented in Table 2. The mean duration for all eight animations was estimated as 4.56 s, which is slightly less than the actual duration. The distribution was tested for normality with a normal QQ plot. The duration perceptions of animations with percentage indicators (Nos. 3 and 6) were estimated as the longest. The identity animation (No. 3) duration was estimated as the shortest.

The subjective feelings of duration and passage of time were evaluated on a five-point Likert scale (1—long, slow; 5—short, fast). The duration and passage of time for the identity and activity animations (Nos. 2 and 7) were perceived as the shortest in duration and the fastest in speed, respectively. Animation 3 (the percentage indicator) was perceived as the longest in terms of duration and subjective feeling of the passage of time.

Perception of interest and aesthetics were evaluated on a five-point Likert scale (1—boring, 5—interesting; 1—low, 5—high). The interest and the aesthetics for animations 2 and 7 were estimated as the highest. The percentage progress indicator was estimated as the lowest in terms of interest and aesthetics. Semantic differential questions (a set of 10 items that covered two subscales) were used to identify the connotative meanings of animation concepts [20]. The participants indicated how much they experienced the effects of each animation on a five-point Likert scale. Affective/emotional factors

were Sad—Happy, Uncomfortable—Comfortable, Tense—At Ease, Anxious—Relaxed, and Unimpressed—Stimulated. Functional–utilitarian factors were Impatient—Patient, Tired—Energetic, Helpless—Powerful, Annoyed—Excited, and Confused—Confident. The affect and the utility of animation 7 were estimated as the highest. The connotative meaning of the percentage progress indicator animation was estimated as the lowest in terms of affect and utility.

**Table 2.** Results of the descriptive statistics for seven QoE features.

| Loading Animation | | Perceived Duration | Perceived Subjective Passage of Time | Perceived Duration | Perceived Interest | Perceived Aesthetics | Affective/ Emotional Factors | Functional– Utilitarian Factors |
|---|---|---|---|---|---|---|---|---|
| | | ms | Likert Scale 1—Slow, 5—Fast | Likert Scale 1—Long, 5—Short | Likert scale 1—Boring, 5—Interesting | Likert scale 1—Low, 5—High | Mean | Mean |
| 1. | Throbber | 4310 | 3.40 | 3.27 | 3.53 | 3.87 | 3.65 | 3.64 |
| 2. | Identity Animation | 4280 | 4.10 | 3.93 | 4.37 | 4.42 | 3.98 | 4.02 |
| 3. | Progress Bar | 5150 | 2.90 | 2.80 | 3.20 | 3.04 | 3.40 | 3.43 |
| 4. | Attribute Animation | 4460 | 3.60 | 3.87 | 3.87 | 3.84 | 3.80 | 3.80 |
| 5. | Circular Indicator | 4330 | 3.90 | 3.67 | 4.13 | 4.1 | 4.09 | 4.07 |
| 6. | Percentage Spinner | 4780 | 3.71 | 3.60 | 3.83 | 3.94 | 3.87 | 3.94 |
| 7. | Activity Animation | 4480 | 4.27 | 3.97 | 4.60 | 4.34 | 4.39 | 4.32 |
| 8. | Text Animation | 4660 | 3.97 | 3.90 | 3.87 | 3.64 | 3.89 | 3.91 |

A one-way repeated measures multivariate analysis of variance (one-way repeated measures MANOVA) was performed to examine whether the eight proposed designs (throbber, identity animation, progress bar, attribute animation, circular indicator, percentage spinner, activity animation, and text animation) differed in four variables of wait-time-animation evaluation (perceived speed, perceived aesthetics, and two semantic differential dimensions: affective and utilitarian). A one-way repeated measures MANOVA was also used to test the differences in multiple dependent variables between the experimental conditions (all participants took part in each experimental condition). The results (Table 3) of the multivariate tests of within-subject effects showed statistically significant differences in wait-time-animation evaluation (in general) between the eight proposed designs (F = 2.763, $p < 0.001$; Wilk's $\Lambda = 0.693$; partial $\eta^2 = 0.09$).

Table 3 contains the results of the tests of within-subject contrasts. As is evident from the table, participants significantly differed in responses for all four variables of wait-time-animation evaluation (perceived duration, perceived aesthetics, and two semantic differential dimensions: affective and utilitarian), depending on the design of each wait-time animation.

The estimated marginal mean values of the four variables of wait-time-animation evaluation for each of the eight proposed designs (throbber, identity animation, progress bar, attribute animation, circular indicator, percentage spinner, activity animation, and text animation) are given in Figure 2.

**Table 3.** Results of the one-way repeated measures MANOVA for the differences of wait-time-animation evaluation depending on design (tests of within-subject contrasts).

| Variables of Wait-Time-Animation Evaluation | F | df | *p* | η² |
|---|---|---|---|---|
| Perceived Speed | 10.039 | 1. 29 | 0.004 | 0.26 |
| Perceived Aesthetics | 8.958 | 1. 29 | 0.006 | 0.24 |
| Affective Dimension | 9.703 | 1. 29 | 0.004 | 0.25 |
| Utilitarian Dimension | 8.202 | 1. 29 | 0.008 | 0.22 |

Note. F = F-value; *p* = *p*-value; df = degrees of freedom; η² = partial eta-squared.

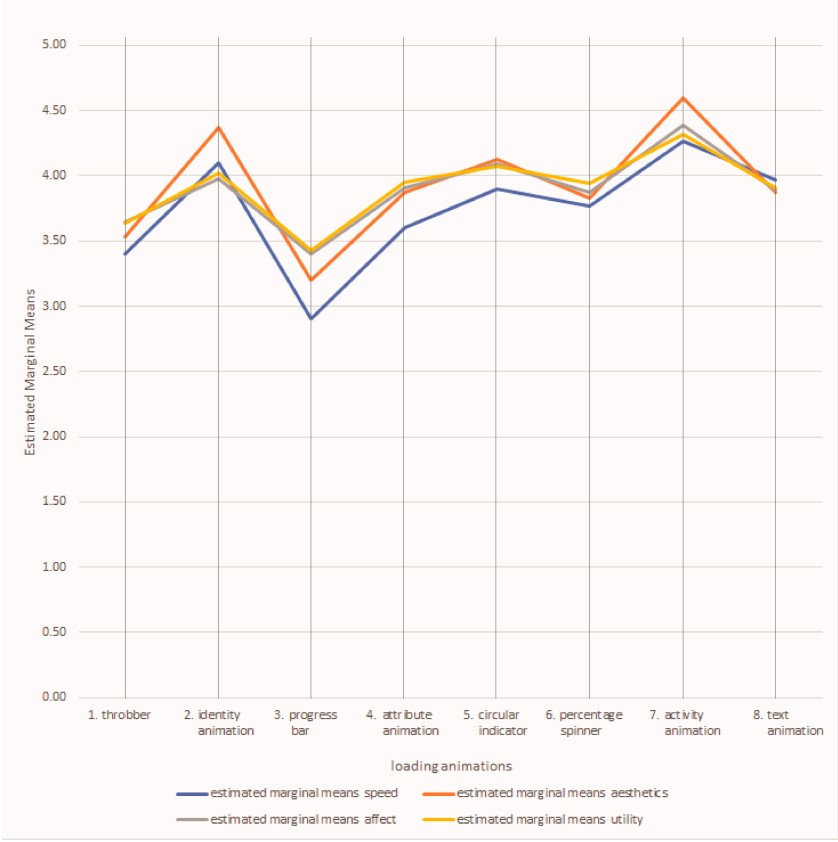

**Figure 2.** The estimated marginal mean values of the four variables of wait-time-animation evaluation for each of the eight proposed designs.

These results confirm the positive effects of semantic meaning, anticipation, and stimulus novelty on subjective duration. Repetitive and abstract loading-animation designs were perceived to last longer and evaluated as less aesthetically pleasing. Since the semantic differential score differences did not follow the same pattern exactly, we can conclude that the differences between the subjective durations of repeated and novel items were related to accuracy of discrimination [53].

### 2.2. Experiment 2—Attention Measure

To check temporal dynamics of distortions in subjective time, e.g., time and space compression related to attention (e.g., saccadic eye movements) to repetitive, semantically meaningless stimuli (circular indicator, No. 3 and progress bar, No. 5) compared to stimuli with semantic meaning (activity animation, No. 7), we conducted an eye-tracking experiment. Previous research has demonstrated that predictable stimuli tend to increase subjective duration [54]. In addition, paying attention can influence perceived duration.

Visual attention movement for three loading animations was measured based on the number of fixations for each. In line with previous research, our hypothesis was that the number of saccadic eye movements would be positively related to perceived duration.

### 2.2.1. Participants

Eleven students and ten faculty staff volunteered as observers, all with normal vision and naïve to both the experimental hypotheses and stimulus parameters.

### 2.2.2. Stimuli and Apparatus

This research was carried out with eye-tracking equipment (Tobii X60) and a 22-inch Lenovo ThinkVision L2251x monitor with a resolution of 1680 × 1050 pixels (with a ratio of 16:10).

### 2.2.3. Procedure

This experiment was conducted under the supervision of an investigator at the User Experience Lab at the Faculty of Graphic Arts, University of Zagreb, in a lightly obscured area with constant light conditions of neutral coloration. The equipment was calibrated, and each examinee was presented with the onscreen display of an app splash screen. He or she had to click on the screen for loading to begin. The order of animations was randomly changed for each examinee. After all three animations were displayed, the examinee was instructed that they would have to compare the durations of the videos (retrospective paradigm). For the comparison, they had to mark the perceived length of each video on a Likert scale from 1 to 10, where 1 was very long and 10 was very short.

### 2.2.4. Results with Discussion

The descriptive fixation statistics for each animation are presented in Table 4. The mean fixation value for repetitive, semantically neutral stimuli (the circular indicator) was the highest; the mean value for the progress bar was the lowest. The results for all three animations were highly correlated, and there was a noticeable difference between the minimal and maximal number of fixations between the observers. The subjective duration-ranking results confirmed the results of Experiment 1: the activity animation was ranked as the shortest and the progress bar as the longest animation. There was no correlation between the number of eye movements and perceived duration. Therefore, our results indicate that subjective duration was not dependent on visual attention or the amount of visual information processed but rather on semantic novelty.

**Table 4.** Fixations in the eye-tracking experiment.

| Stimulus | Mean | Max | Min | Sum | Stdev |
|---|---|---|---|---|---|
| Circular Indicator (No. 3) | 11.36 | 22 | 8 | 250 | 3.39 |
| Progress Bar (No. 5) | 9.36 | 15 | 5 | 206 | 2.77 |
| Activity Animation (No. 7) | 9.82 | 20 | 5 | 216 | 3.5 |

## 3. Discussion

The compilation of the research findings from the psychology/neuroscience and computer interaction studies was the basis for this study's time-duration influence-factor determination. We related those factors to the ways that users perceive, understand, and experience visual stimuli. Visual information processing was the reference for loading-animation interpretation analysis. These experimental results are in line with previous findings, which state that loading-animation design and semantic meaning have the strongest influences on wait-time perception and, consequently, on system-performance perception. The appraisal theory has indicated the connection between wait-time-animation semantic connotations and participants' judgments of the pleasantness of the experiences. Through design and communication of narrative sequences related to the activity and the identity

of an application, a designer provokes meaning. Such an animation makes time seem to pass more quickly and the waiting period much more pleasant, especially if it is related to the application's purpose and fosters a sense of anticipation (in this case, a cup of coffee). Activity animation, followed by identity animation, was evaluated as the most interesting and aesthetically pleasing; its connotative meaning was evaluated as the most affective and utilitarian. The participants' subjective passage of time and duration estimations for those animations were perceived as the shortest. Those distortions in perceived duration were caused by semantic novelty rather than attentional orientation.

## 4. Conclusions

We can conclude that mobile-application loading-design decisions are closely related to wait-time experience and perception of speed. Loading animations designed to provoke meanings and emotions related to the users' goals trigger immediate psychological processes, thereby shortening perception of duration. However, due to their complexity, these animations tend to have higher weights and use more processing power, which decrease perceived loading speed. On the other hand, the predictable designs and repeated actions of generic indicators do not provide meaning, interest, or affection, but their advantages certainly lie in small file size, fewer skills and less amount of work required for their design and programming, and less processing power needed for their smooth running. Therefore, animation scenarios and graphics must be carefully worked out to ensure that appropriate animation techniques and optimized image sizes are used without compromise of perceptual fluency, pleasantness, or meaning of the experience. From an implementation viewpoint, the workflow of JSON-based animations, which enables smaller file sizes, instant preview, and testing on mobile devices, is a promising new technology.

The research conducted here has certain limitations that need to be addressed in future work. This experiment could be redesigned as a between-subjects study with more participants. There are influencing factors whose effects on QoE (quality of experience) need to be further investigated, such as context, mobile device, purpose of waiting, user motivation, and urgency. In addition, metrics relevant to how users perceive performance, such as perceived loading speed, loading responsiveness, visual stability, and smoothness, should be included.

**Author Contributions:** Conceptualization, J.P.; methodology, J.P. and J.D.; validation, L.M.; formal analysis, L.M.; investigation, V.K.; resources, J.P.; visualization, V.K. All authors have read and agreed to the published version of the manuscript.

**Funding:** This research received no external funding.

**Informed Consent Statement:** Informed consent was obtained from all subjects involved in this study.

**Data Availability Statement:** Not applicable.

**Conflicts of Interest:** The authors declare no conflict of interest.

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
