# Peer review of "Mobile-Application Loading-Animation Design and Implementation Optimization"

_applsci, doi:10.3390/app13020865_

Round 1
Author Response
Dear Reviewer,
We wanted to express my sincere gratitude for taking the time to review the article. Your insights and suggestions have been invaluable in helping us improve our work's quality and clarity.
We sincerely appreciate the thought and care you put into your review, and we are grateful for the opportunity to address your comments and suggestions. We have made corrections according to each and every one of your remarks—paragraph 3.1. is incorporated into the Introduction and Conclusion has been reframed.
Your feedback has been invaluable in helping us to shape and refine our research, and we are confident that the final version of the article will be stronger as a result.
Thank you again for your time and expertise.

Reviewer 2 Report
In this study, animation design and implementation for mobile platforms were presented. Topic is interesting but the paper is not quite good for this journal. The abstract is insufficiently informative. Comments are given below:
1. The abstract is insufficiently informative. State of the art of the study is not clearly defined. (differences between other methods and your performance of the system with percentages should be given).
2. There is no achievement in this part of the study.
3. Literature search given in a mass, some references such as [1-5], [6-9] are only given as groups, there is no detailed explanation about these references.
4. Reading introduction and understanding the literature study are too hard for readers.
5. There is no block diagram or pseudocode are given for readers. Works are too limited. Section 2 should be enlarged using meaningful study details.
6. There is no Figure except duration and aesthetics in the paper about the work, this kind of papers generally too boring for authors.
7. Differences between other methods and your performance of the system with percentages should be given in discussion.
8. Performance evaluation is limitedly given in section 3, but these evaluations should be enlarged with literature.
9. Your future aspects should be given in details in Conclusion.
Author Response
Dear Reviewer,
We express our sincere gratitude for taking the time to review our article. Your thorough and thoughtful comments have been invaluable in helping us improve the quality and clarity of my work.
We particularly appreciate your constructive feedback, which has helped us to address weaknesses and clarify misunderstandings. Your expertise and insights have been invaluable in helping us to refine concluding arguments and better communicate our ideas:
- The abstract has been reframed
- References to other methods and explanations of references [1-5]and [6-9] have been added
- A Block diagram explaining the research process has been created
- Differences between other methods have been discussed
- The conclusion has been reframed and future aspects added
We are grateful for the opportunity to have benefited from your knowledge and experience, and we hope that the revised article will meet your expectations.
Once again, thank you for your invaluable contribution to the review process.

Round 2
Reviewer 2 Report
.